# Radiation-Induced Bystander Effect: Loss of Radioprotective Capacity of Rosmarinic Acid In Vivo and In Vitro

**DOI:** 10.3390/antiox10020231

**Published:** 2021-02-03

**Authors:** Amparo Olivares, Miguel Alcaraz-Saura, Daniel Gyingiri Achel, Juan de Dios Berná-Mestre, Miguel Alcaraz

**Affiliations:** 1Radiology and Physical Medicine Department, School of Medicine, University of Murcia, 30100 Murcia, Spain; amparo.o.r@um.es (A.O.); Miguel.Alcaraz@um.es (M.A.-S.); juandeberna@um.es (J.d.D.B.-M.); 2Applied Radiation Biology Centre, Radiological and Medical Sciences Research Institute, Ghana Atomic Energy Commission, Legon-Accra GE-257-0465, Ghana; d.achel@gaecgh.org

**Keywords:** radiation effects, bystander, radioprotectors, micronucleus, PNT2, B16F10, TRAMPC1, rosmarinic acid

## Abstract

In radiation oncology, the modulation of the bystander effect is a target both for the destruction of tumor cells and to protect healthy cells. With this objective, we determine whether the radioprotective capacity of rosmarinic acid (RA) can affect the intensity of these effects. Genoprotective capacity was obtained by determining the micronuclei frequencies in in vivo and in vitro assays and the cell survival was determined by the (3-(4,5-dimethylthiazol-2-yl)-2,5-diphenyltetrazolium bromide assay) (MTT) assay in three cell lines (PNT2, TRAMPC1 and B16F10), both in direct exposure to X-rays and after the production of radiation-induced bystander effect. The administration of RA in irradiated cells produced a decrease in the frequency of micronuclei both in vivo and in vitro, and an increase in cell survival, as expression of its radioprotective effect (*p* < 0.001) attributable to its ability to scavenge radio-induced free radicals (ROS). However, RA does not achieve any modification in the animals receiving serum or in the cultures treated with the irradiated medium, which expresses an absence of radioprotective capacity. The results suggest that ROS participates in the formation of signals in directly irradiated cells, but only certain subtypes of ROS, the cytotoxic products of lipid peroxidation, participate in the creation of lesions in recipient cells.

## 1. Introduction

The traditional concept of radiobiology is based on the “target” theory, which assumes that all biological effects on irradiated cells are the result of direct cell damage produced by ionizing radiation [1]. However, during the last two decades, this classic paradigm has changed due to the discovery of non-directed effects (of target) of ionizing radiation [1,2,3,4]. These undirected effects of ionizing radiation include the radiation-induced bystander (Bys) effect: the ability of target cells injured by ionizing radiation to induce secondary biological changes in non-irradiated cell recipients [5,6,7,8,9,10].

The actual mechanisms underlying the radiation-induced Bys effect are not yet completely clarified, although many of the factors involved in it are currently being unraveled. Since the beginning of the investigation of this phenomenon, it was assumed that the induction factors of the Bys effect are produced directly in irradiated cells and transmitted to non-irradiated cells, causing them certain biological changes [11]. The Bys effect has been reported to be manifested by changes in the expression of certain genes and protein synthesis, cell death (apoptosis or necrosis), proliferation, differentiation, cell transformation, adaptive response, aging, free radical formation, induction of mutations genetics, chromosomal aberrations, sister-chromatid exchanges (SCE), and micronuclei [12,13,14]. It can be assumed that the initial trigger for Bys responses is a nonspecific activation of a redox-sensitive signaling pathway through mitochondrial induction of reactive nitrogen and oxygen species (ROS/RNS) [15]. ROS induced by ionizing radiation are considered to be the initiators, and nitric oxide the effector, that activates the Bys process through a signal transduction pathway [16]. Transforming growth factor beta (TGF-beta) protein has been described as an important agent in signal activation and in amplifying the response to oxidative stress [17,18,19], which together with other inflammatory cytokines appear to increase in irradiated tissues [20,21] and contribute to injury induced by ionizing radiation and also in Radiation-Induced Bystander Effect (RIBE) [1,22,23].

Rosmarinic acid (RA), present in many plant species, is a common ester derived from caffeic acid and (R)-(+)-3-(3,4-dihydroxyphenyl) lactic, that shows numerous biological activities of medicinal interest [24,25,26]. RA has been demonstrated to possess important radioprotective and genoprotective capabilities in different tumor versus normal cell lines by blocking direct damage induced by ionizing radiation (X-rays and gamma rays) [27,28,29,30,31,32,33,34] and even by ultraviolet radiation [29]. This radioprotective capacity of RA has been attributed to its antioxidant capacity measured by its capability of eliminating free radicals induced by the action of ionizing radiation and directly affecting irradiated cells (“target” theory) [29,34].

From a clinical point of view, RIBE modulation is a desirable target in radiotherapy due to its genotoxic potential that would allow the destruction of tumor cells and protect healthy cells that are in the same irradiation field. Radioprotective substances have been developed that protect against injuries caused by direct exposure to ionizing radiation, and other substances have also been developed that could reduce damage to receptor cells [1,28]. In radiation oncology, radioprotective substances could protect the normal tissue surrounding tumors and ameliorate the side effects of therapy, but, in this setting, care must be taken not to diminish the destruction of tumor cells. Therefore, a complex understanding of passerby signaling pathways and potential molecular targets is crucial [1,35,36].

In this work, we propose to evaluate the genoprotective and radioprotective effects of RA on lesions induced by ionizing radiation in an indirect way, through radiation-induced Bys effect, which would allow us to evaluate its antioxidant ability to protect against these lesions.

## 2. Materials and Methods

### 2.1. Chemicals and Reagents

Rosmarinic acid (95%) (Figure 1) was obtained from Extrasynthese (Genay, France) and dimethyl sulfoxide (DMSO) was obtained from Merck (Darmstadt, Germany). Roswell Park Memorial Institute Medium (RPMI) 1640, F10, Phytohaemagglutinin (PHA), cytochalasin B, streptomycin, penicillin, phosphate buffered saline (PBS), methanol, heparin, sodium chloride, sodium bicarbonate, RNase A from bovine pancreas and propidium iodide 1.0 mg/mL solution were obtained from Sigma-Aldrich Chemicals S.A (Madrid, Spain). Fetal bovine serum (FBS) was obtained from Gibco (Life Technologies S.A., Madrid, Spain). Fluorescein isothiocyanate (FITC)-conjugated rat anti-mouse CD71 transferrin receptor antibody was from Southern Biotech (Birmingham, USA).

### 2.2. Genototoxic Effect

#### 2.2.1. In Vivo Micronucleus Assay

In the in vivo experiments, 12-week-old male Swiss mice distributed in groups of 6 animals for each group with a weight ranging from 27 to 35 g were used. All solutions were prepared daily and the RA (rosmarinic acid (95%), Extrasynthese, Genay, France) (Figure 1) was dissolved to a concentration of 0.2% in their drinking water during one week prior to X-ray exposure.

##### Micronucleus Assay in Mouse Bone Marrow (Micronuclei in Polychromatic Erythrocytes (MNPCEs))

The in vivo micronucleus test was conducted on mouse bone marrow, as previously described [37]. After twenty-four hours of X-ray exposure, the frequency of micronucleated polychromatic erythrocytes (MNPCEs) among 1000 polychromatic erythrocytes (PCEs) per mouse were evaluated by three specialists in a double-blind study. In the RA groups, the animals were given drinking water containing 0.2% RA ad libitum for a period of two weeks before exposure to X-rays. To show that RA is not toxic to the cells used, the frequencies of normochromatic erythrocytes and total erythrocytes were also assessed in each animal.

##### Flow Cytometry for Micronuclei in Reticulocytes (MNRET)

Flow cytometry-based micronuclei (MN) analysis in reticulocytes (RET) was performed as described by Balmus et al. [38]. A Becton Dickinson LSR Fortessa Flow cytometer with high-throughput sampler option, capable of handling multi-tier-well plates (96-well plates) and equipped with 488 nm blue and 561 nm green lasers was employed (BD Bio-sciencies, San Jose, CA, USA). Fluorescence emitted from FITC-conjugated anti-CD71 and propidium iodide (PI) were collected through a 530/30 nm and a 610/20 nm band-pass filter, having been excited by the blue 488 nm and green 561 nm lasers, respectively. A high-throughput sampler was set to mix each sample 2–4 times, whereupon 196,000 events per sample were acquired at a rate of 1000–2000 events per second. The preparation of single fluorochrome and unstained controls from surplus sample was used to set-up the regional gates for the analysis of the erythrocyte cell population. Approximately 196,000 events per animal were scored to obtain the frequency of micronucleated reticulocytes in the blood (MN-RET), which was calculated using absolute events from the different quadrants obtained from the software using the following formula: % MN-RET = [MN-RET/(RET + MN-RET)] × 100.

#### 2.2.2. In Vitro Micronucleus Assay

##### Micronucleus Test In Vitro (Cytokinesis-Block Micronucleus Assay (CBMN))

For the in vitro genotoxicity test, 20 mL of venous blood was obtained from the elbow flexure from three apparently healthy human donors after their informed consent, which were collected in heparinized tubes for the performance of the human lymphocyte micronucleus assay with cytokinesis-block (CBMN), as previously described [39] and adapted by the International Atomic Energy Agency for the screening of genoprotective substances against damage induced by ionizing radiation [40]. In the RA groups, 25 microliters (µL) of a 25 µM concentration of RA dissolved in PBS were added to 2 mL of heparinized human blood immediately prior to X-ray exposure. The number of micronuclei in at least 3000 CB cells for each treatment was determined by three specialists who analyzed the slides using light microscopes in a double-blind study.

### 2.3. MTT Cytotoxicity Assay

#### 2.3.1. Cell Culture

We selected three cell lines based on their degree of radiosensitivity. On the one hand, melanoma cells are characterized by high radio-resistance [41], which is why we used murine metastatic melanoma cells B16F10, which were kindly donated by Dr. Hearing from the National Cancer Institute (Bethesda, MA, USA). In contrast, human prostate cells are traditionally considered radiosensitive [41], so we selected human prostate epithelial PNT2 cells which were obtained from the European Collection of Cell Cultures (ECACC), Health Protection Agency Culture Collection (catalog n: 95012613; UK). Finally, to examine the tumor influence on these radiosensitive cells, we also used transgenic adenocarcinoma of mouse prostate (TRAMP-C1) cells, which were obtained from the American Type Culture Collection (catalog n: CRL-2730, USA). The PNT2 cells were cultured in RPMI-1640 medium supplemented with 10% fetal bovine serum, and glutamine (2 mM) containing streptomycin 100 μg/mL and 100 IU/mL penicillin. Both TRAMP-C1 and B16F10 cells were cultured in Dulbecco’s Modified Eagle’s Medium and Nutrient Mixture-F2 Ham (DMEM)/F12K (1:1). Both mediums contained 10% fetal bovine serum (Gibco, BRL, Louisville, KY, USA) and 5% penicillin/streptomycin. The cell cultures were maintained at 37 °C and a relative humidity of 95%, in an atmosphere of 5% CO_2_. Rosmarinic acid was administered by dissolving it in phosphate buffered saline (PBS) to a concentration of 25 µM and consistently adding 25 μL of the solution to each well immediately before X-ray irradiation (10 mn). At all times, the cell cultures were kept free of mycoplasma spp.

#### 2.3.2. MTT Assays of Irradiated Cells

To evaluate the effects of IR on the viability and survival of PNT2, TRAMP-C1 and B16F10 cells, the MTT assay lasting 24 and 48 h was employed to evaluate cell survival and proliferation, respectively [42,43]. For the PNT2 and TRAMPC1 cells, 3200 cells/well and for B10F16, 2500 cells/well were established as optimal seeding densities. Absorbance readings were captured using a Multiskan MCC/340P spectrophotometer at a wavelength of 570 nm for the test and 690 nm as a reference wavelength. The negative control wells were used for the baseline zero. Each experiment was repeated eight times.

### 2.4. Models of Radiation-Induced Bystander Effects (RIBE)

For the in vivo micronucleus assay, 0.2 mL serum from irradiated animals was administered by intraperitoneal puncture to each animal to elicit RIBE effects. In the BysRA group, the animals were given drinking water containing 0.2% RA ad libitum for a period of two weeks. In the mouse bone marrow micronucleus assay, bone marrow was harvested from the medullary canal of the femur 24 h after exposure to X-ray irradiation. In the MN-RET assay by flow cytometry, blood samples were obtained by intracardiac puncture 48 h post-irradiation (Figure 2a).

In the vitro micronucleus assay (CBMN), 0.1 mL serum from irradiated blood was added to the blood samples before commencement of the cytochalasin B block micronucleus test with human lymphocytes. In the BysRA group, 25 microliters (µL) of a 25 micromolar (µM) solution of RA dissolved in PBS had previously been added (Figure 2b).

In the MTT assays, we used the protocol of “medium transfer” from irradiated to non-irradiated cells. For this protocol, PNT2, TRAMPC1 and B16F10 cells were seeded to confluence. At the time of starting X-ray exposure, we changed the medium to fresh medium in all cells (non-irradiated cells, non-irradiated cells + RA, irradiated cells and irradiated cells + RA); 4 h after irradiation, this medium was collected, where the factors that can induce Bys effects are found. This medium was subsequently centrifuged (200 g) and transferred to 96-well plates containing transient PNT2, TRAMPC1 and B16F10 cells in confluence. In the BysRA group, 25 microliters (µL) of a 25 micromolar (µM) solution of RA dissolved in PBS had been previously added. Bys cells were kept in culture with this medium for 48 h and analyzed as described in the MTT assays for the study of the direct effect induced by X-rays (Figure 2c).

### 2.5. Irradiation

An X-ray producing equipment (Andrex SMART 200E, Yxlon International, Hamburg, Germany) with the following operational features was used: 200 kV, 4.5 mA, filtration of 2.5 mm of Al and dose rate of 1.3 cGy/s at a focus object distance (FOD) of 35 cm. Experiments were performed at room temperature. For in vivo genotoxic evaluations, conscious bur immobilized animals were total-body irradiated to 50 cGy X-rays in order to evaluate MNPCEs frequencies in bone marrow and 2 Gy to determine MN-RET by flow cytometry. In the in vitro genotoxicity assay, whole human blood samples were exposed to 2 Gy X-rays for micronuclei assessment in cytochalasin B (CBMN) blocked cells. In the MTT assay, cells grown in microplates were irradiated to 20 Gy of X-rays. Radiation doses delivered during the entire period of experimentation were continuously monitored inside the X-ray cabin by means of UNIDOS^®^ Universal Dosimeter with PTW Farme^®^ ionization chambers TW30010 (PTW-Freiburg, Freiburg, Germany). The final radiation dose was confirmed by means of thermo-luminescent dosimeters (TLDs) (GR-200^®^; Conqueror Electronics Technology Co Ltd., Shenzhen, China).

### 2.6. Statistical Analysis

An analysis of the frequency of micronuclei gives an indication of the genotoxic nature of the substance studied. In this study, the degree of dependence and correlation between variables was assessed using analysis of variance (ANOVA) complemented by contrasting their means. Quantitative means were compared by regression and linear correlation analysis. To determine protection factors (PF) with regards to the frequency of occurrence of MN, the formula proposed by Sarma and Kesavan [44] was invoked regarding the: PF (%) = (Fcontrol − Ftreated/Fcontrol) × 100, where Fcontrol is the frequency of micronuclei in the irradiated control samples, and Ftreated is the frequency of micronuclei in the samples treated with RA and irradiated.

For the cell survival analysis which portrays cytotoxicity of the test materials, an analysis of variance (ANOVA) of repeated measures was conducted to compare percentages of surviving cells in the cultures, this was complemented by least significant difference analyses to contrast the differences between any pairs and means. In the cytotoxicity analysis, we modified formula proposed by Sarma and Kesavan [44] to adapt it to the cell survival cultures exposed to 20 Gy and incubated over a period of 48 h, as follows: PF (%) = (Mcontrol − Mtreated/Mcontrol) × 100, where Mcontrol is the mortality of the irradiated control cells, and Mtreated is the mortality of the cells treated with RA and irradiated.

## 3. Results

### 3.1. Genototoxic Effect

#### 3.1.1. In Vivo Micronucleus Assay

##### Micronucleus Assay in Mouse Bone Marrow (MNPCEs)

The basal frequency determined in these animals was 3 MN/1000 PCEs, which does not show any significant differences with the frequency observed in the animals treated with RA, and depicts an absence of genotoxic effect of RA.

Exposure to 50 cGy X-rays produced a significant increase in the frequency of appearance of MN (*p* < 0.001), which is an expression of the genotoxic damage induced by ionizing radiation, while the administration of RA before irradiation produced, on the contrary, a decrease (*p* < 0.001) in the frequency of appearance of micronuclei with a protection factor (PF) of 61%, which expresses antimutagenic and genoprotective capacities of RA against damage induced by ionizing radiation (Figure 3).

In animals treated with 25 microliters of serum from irradiated animals (Bys), a significant increase in the frequency of MN was determined compared to non-irradiated control animals (*p* < 0.01), which expresses the existence of genotoxic damage induced by the administration of serum from irradiated animals. However, the administration of RA before the intraperitoneal injection of the irradiated serum did not show differences in the frequency of MN, which expresses a lack of RA effect in these Bys animals. Therefore, the following order can be established moving from lower to higher frequency of appearance of MN in PCEs cells: C = RA < Bys = BysRA = RAi < Ci (*p* < 0.001) (Figure 3).

##### Flow Cytometry for Micronuclei (MN) in Reticulocytes (RET)

The basal frequency determined in the control animals by flow cytometry was 1.5 MNRET/100 RET (1.5%), which does not present significant differences with the frequency observed in the animals treated with RA (1.8%), showing an absence of genotoxic effect to RA in these animals. Exposure to 2 Gy of X-rays produced a significant increase in the frequency of appearance of MN (7.8%) (*p* < 0.001), an expression of the genotoxic damage induced by X-rays, while the administration of RA before irradiation produced, on the contrary, a decrease (6.4%) (*p* < 0.01) in the frequency of appearance of micronuclei in reticulocytes, with a PF of 18%, which expresses an antimutagenic and genoprotective capacity of RA against damage induced by ionizing radiation (Figure 4). In animals treated with 25 microliters of serum from animals irradiated with 2 Gy (Bys), a significant increase in the frequency of MN was determined compared to non-irradiated control animals (2.6%) (*p* < 0.01), which shows the existence of genotoxic damage induced by the administration of serum from irradiated animals (Bys). However, the administration of RA before intraperitoneal injection of the serum from irradiated animals showed no significant differences, expressing a lack of effect of RA in these Bys animals. Therefore, the following order of frequency of MN appearance in blood reticulocytes was established, from lower to higher frequency: C = RA < Bys = BysRA < RAi < Ci (*p* < 0.001) (Figure 4).

#### 3.1.2. In Vitro Micronucleus Assay

##### Micronucleus Test In Vitro (CBMN)

The baseline micronuclei frequency determined in irradiated cytochalasin B-blocked human lymphocytes was 10 MNCB/500 CB, which does not present significant differences with the frequency observed in cultures treated with RA, showing an absence of genotoxic effect of RA. Exposure to 2 Gy of X-rays produced a significant increase in the frequency of appearance of MN (27 MNCB/500 CB) (*p* < 0.001), an expression of genotoxic and mutagenic damage induced by ionizing radiation, while the administration of RA before irradiation produced, on the contrary, a decrease (*p* < 0.001) in the frequency of MN appearance, with a PF of 55%, which expresses an antimutagenic and genoprotective capacity of RA against damage induced by ionizing radiation (Figure 5). In human lymphocyte cultures treated with 25 microliters of serum from 2 Gy irradiated blood, a significant increase in the frequency of MN (*p* < 0.01) was observed, which shows the existence of genotoxic damage induced by serum obtained from irradiated blood. However, the administration of RA under these conditions did not show differences, which expresses an absence of effect of RA in these Bys + RA cells. Therefore, from lower to higher frequency of appearance of MN in CB cells, the following order was established: C = RA = RAi < Bys = BysRA < Ci (*p* < 0.001) (Figure 5).

### 3.2. MTT Cytotoxicity Assay

#### 3.2.1. PNT2 Cells

The administration of RA did not modify the cell survival of PNT2 cells, showing an absence of toxicity of RA in these cells incubated for 48 h and at the concentration tested. In PNT2 cells, exposure to 20 Gy of X-rays produced a significant decrease in cell survival (*p* < 0.001), an expression of radiation-induced cytotoxicity, while the administration of RA in these cell cultures before irradiation produced, on the contrary, a significant increase in cell survival (*p* < 0.001), with a PF of 88%, which expresses a radioprotective capacity of RA (Figure 6). Cell survival of PNT2 cells treated with centrifuged cell culture medium from PNT2 cell cultures irradiated with 2 Gy (Bys) caused a significant decrease in cell survival (*p* < 0.01), demonstrating cytotoxicity induced by the irradiated culture medium. However, the administration of RA on these cells did not show significant differences, which expresses a lack of effect of RA in these Bys cultures. Therefore, from higher to lower cell survival, the following order was established: C = RA = RAi > Bys = BysRA > Ci (*p* < 0.001) (Figure 6).

#### 3.2.2. TRAMPC1 Cells

The administration of RA does not portray significant differences in the survival of TRAMPC1 cells, showing an absence of toxicity of RA in these cells incubated for 48 h and at the concentrations tested. In TRAMPC1 cells, exposure to 20 Gy of X-rays was noted to produce a significant decrease in cell survival (*p* < 0.001) which is an expression of the radiation-induced cytotoxicity, while the administration of RA produced, on the contrary, a significant increase in cell survival (*p* < 0.001), with a PF of 90%, which expresses a radioprotective effect of RA against damage induced by ionizing radiation (Figure 7).

Cell survival of TRAMPC1 cells treated with clarified cell culture medium obtained from TRAMPC1 cell cultures irradiated to 2 Gy (Bys) provoked a significant decrease in cell survival (*p* < 0.01), which is an indication of cytotoxicity induced by the irradiated culture medium. However, the administration of RA to these cells did not show any significant differences, which expresses a lack of effect of RA in the Bys cultures Therefore, making a ranking from highest to lowest cell survival, the following order was established: C = RA = RAi > Bys = BysRA > Ci (*p* < 0.001) (Figure 7).

#### 3.2.3. B16F10 Cells

Administration of RA did not portray significant differences in the survival of B16F10 cells, showing an absence of toxicity of RA in these cells incubated for 48 h and at the concentration tested. In B16F10 cells, exposure to 20 Gy of X-rays produced a significant decrease in cell survival (*p* < 0.001), an expression of radiation-induced cytotoxicity. However, the administration of RA before irradiation produced a significant decrease in cell survival (*p* < 0.001), which expresses a radio-sensitizing effect of RA, which significantly increases radiation-induced cell death (Figure 8). Cell survival of B16F10 cells treated with clarified cell culture medium obtained from B16F10 cell cultures exposed to 2 Gy of irradiation (Bys) caused a significant decrease in cell survival (*p* < 0.01), which demonstrates the existence of a cytotoxicity induced by the irradiated culture medium. However, the administration of RA to these cells did not lead to significant differences in cell death, which expresses a lack of effect of RA in the Bys cultures. Therefore, when making a rank order in cell survival from highest to lowest, the following was established: C = RA > Bys = BysRA > Ci > RAi (*p* < 0.001) (Figure 8).

## 4. Discussion

The molecular mechanisms involved in the Bys response seem to show the existence of a common mechanism through which cell cytotoxic and genotoxic stress can occur. Similar RIBE-like Bys produced by other genotoxic stressors such as chemotherapy, tumors, photodynamic stress, contamination with heavy metals or induced by microbiomes, nanoparticles and even heat have been described [1,45].

A unifying model system for genotoxic stress responses has been proposed in Radiobiology [1,46] that describes the initial release of free radicals whose release is triggered by a variety of events, including ionizing radiation. These free radicals are then channeled into a network of redox-sensitive signaling pathways. It is believed that redox signaling mechanisms are fundamental for the propagation of the bystander effects [47] initiated by a generation of ROS, such as those induced by ionizing radiation. ROS induced by ionizing radiation are considered to be the initiators, and nitric oxide the effector, that activate the Bys process through a signal transduction pathway [16]. Mitochondria are an important source of ROS generation, and provide a center for redox activities and control of cellular distress [48,49]. TGF-beta protein has been described as an important agent in signal activation and in amplifying the response to oxidative stress [17,18,19]. Both TGF-β and other inflammatory cytokines appear to increase in irradiated tissues [20,21] and contribute to injury induced by ionizing radiation and also in RIBE [1,22,23]. In addition, TGF-β-induced apoptosis [1,50] is associated with the transition to mesenchymal epithelium [50,51], with an increase in metastasis [52] and even modifying the immune response against cancer [53,54]. It can be assumed that the initial trigger for Bys responses is a nonspecific activation of a redox-sensitive signaling pathways through mitochondrial induction of ROS/RNS [1]. The increase and diffusion of these stress signals result in the induction of DNA damage, the increase of apoptosis and the diminution of cell survival in neighboring tissues, as well as the production of a generalized inflammatory response. This genotoxic stress induced by the redox signaling response could be considered as a nonspecific stress response that could be triggered by different events (ionizing radiation, chemotherapy, photodynamic stress, tumors) that modify the cellular redox balance and disrupt mitochondrial function [1].

In the “target” theory, exposure to ionizing radiation increases cellular oxidation processes causing molecular oxygen to give rise to superoxide radicals (O^●^_2_), hydrogen peroxide and hydroxyl radicals (^●^OH), successively [41,55]. Of special interest is the intense generation of hydroxyl radicals since they are presently considered to have the highest cytotoxic capacity, an adverse effect that can possibly be mitigated by the administration of radioprotective substances such as RA. These hydroxyl radicals have high chemical reactivity and an estimated half-life of 10^−9^ s, which in practice implies an immediate reaction close to the place where they are generated. However, when the generation of these hydroxyl radicals is massive, they interact with the cellular phospholipoid structures, inducing the lipoperoxidation processes and gradually producing lipoperoxy radicals (R^●^, RO^●^, RR^●^, ROO^●^, ROOR^●^). An escalation of these free radicals’ accumulation over time increases cellular genotoxic and cytotoxic effects, in what has been considered a delayed secondary reaction produced by ionizing radiation, which prolongs the cytotoxic effect for at least 24 h after the exposure to ionizing radiation has ended. On the other hand, this increase in lipid peroxidation increases lipoxygenase, cycloxygenase and phospholipase activities, which leads to an increase in the secretion of lysosomal enzymes and the release of arachidonic acid from cell membranes, that also increases the intensity of cellular inflammatory response. Through this delayed reaction of lipoperoxy radicals, ionizing radiation can induce effects at some distance from the place where the exposure to ionizing radiation initially occurred and for a much longer time [28,41,55,56] (Figure 9).

We used three complementary cytogenetic assays to evaluate the mutagenic/genotoxic capacity of ionizing radiation by determining the frequency of appearance of micronuclei in cells and animals exposed to ionizing radiation. Similarly, a reduction in the frequency of occurrence of these micronuclei makes it possible to determine the protection capacity (antimutagenic effect) of RA against chromosomal damage induced by ionizing radiation. Since there is no single test that is capable of detecting all genotoxic end points, the complementary use of several genotoxicity tests is recommended. The homogeneity of results allows for the assessment of the mutagenic capacities of ionizing radiation, irradiated culture medium and irradiated blood serum, as well as the possible antimutagenic capacity of RA. In this study, we applied the most widely used micronucleus assays to determine the genotoxic capacity of ionizing radiation both in vitro and in vivo, which are considered universally validated, technologically accessible and useful for evaluating genetic instability induced by genotoxic agents and especially in screening for radioprotective substances [40]. In the in vitro test on irradiated human lymphocytes blocked with cytochalasin B, the micronucleus, a chromosome fragment that remains outside the mitotic spindle after the first cell division, is observed as a cytoplasmic body with nuclear characteristics that correspond to genetic material not incorporated correctly to daughter cells. This reflects a chromosomal aberration produced by chromosomal breaks, by errors during DNA replication and subsequent cell division and/or by exposure to genotoxic agents or substances [39]. In the in vivo test, the absorption and bioavailability of the physical and chemical agents tested may alter the expected results. In these in vivo tests, the micronucleus is also a chromosomal fragment with nuclear characteristics that remains inside the polychromatophilic erythroblasts of bone marrow (mouse bone marrow). It may also remain inside the circulating reticulocytes in the blood, showing an inexplicable delay in its elimination from the cell as an expression of the delay in cell maturation and proportional to the chromosomal damage induced by a chemical or physical agent [37,38].

In this sense, our genotoxicity tests show a high mutagenic capacity, both X-rays with which was used to directly irradiate the cells and animals studied, as well as the irradiated culture medium and the serum obtained from irradiated animals. However, we observed significant differences between the two since the serum from irradiated animals and the irradiated culture medium expressed significantly lower mutagenic capacities than direct exposure to X-rays.

On the other hand, the reduction of MN after the administration of RA allows for the quantification of the antimutagenic capacity of RA compared to direct exposure to X-rays, while the lack of effect on the frequency of appearance of MN in all the tests carried out expresses the absence of genoprotective capacity of RA in cells that receive Bys signals from irradiated serum or culture medium.

Under these conditions of cellular oxidative stress, the administration of RA achieves a significant antimutagenic capacity, reducing the chromosomal damage induced by X-rays. In general, this capacity depends on the high absolute reactivity towards the different radicals and the high stability of the intermediate radicals formed, which are related to the molecular structure of the substance, its physical properties and bioavailability of RA [27,28,29,30]. It has been established that the capacity of RA to scavenge hydroxyl radicals is primarily due to a combination of its ring system, which is conjugated to double bonds in the polyphenolic skeleton, pre-dominantly the o-dihydroy-phenol or catechol structure. To this end, the existence of two catechol groups in RA which are conjugated to a carboxylic acid functional group enhances its antioxidant activity in an aqueous medium (Figure 1). This radioprotective capacity of RA has been attributed to its antioxidant capacity measured by its capability of eliminating free radicals induced by the action of ionizing radiation and directly affecting irradiated cells (“target” theory) [28,29,34].

Similarly, cell survival increased in PNT2 and TRAMPC1 cells, which is an expression of the radioprotective capacity of RA similar to previously described results, although higher doses of X-rays (20 Gy) were used in this study [27,28,29,30,31,32,33,34,42,43]. On the contrary, in B16F10 melanoma cells, a paradoxical radio-sensitizing effect of RA was confirmed, which produced a greater decrease in cell survival than in the previously described results with 10 Gy of X-rays [42,43]. This has been attributed to the effect of RA on intracellular glutathione, which is less available to eliminate free radicals induced by ionizing radiation, possibly due to it being consumed to form pheomelanin during melanogenesis [34].

A limitation of the study is the exact determination of cell survival using the MTT technique. Currently, the clonogenic assay and the annexin V/Propidium Iodide (V-PI) apoptosis assay are the standard techniques for evaluating cell viability in ionizing radiation treatments. However, the comparative analysis of the results obtained with the MTT test makes it possible to assess the effect of RA in both directly irradiated cells and in Bys cells.

The molecular mechanisms that initiate and propagate Bys responses have been well-researched and understood. On the contrary, there are still large gaps in our understanding of Bys signal processing in receptor cells, which are essential to establish the degree of radiosensitivity to genotoxic stress of these cells and, therefore, do not yet allow us to find strategies of cellular radioprotection [1]. In this study, we have tried to confirm that these genoprotective and radioprotective capacities of RA, which were determined in cells directly exposed to ionizing radiation, are also produced in these recipient Bys cells. This could demonstrate that the antioxidant capacity of RA could also eliminate ROS transmitted by the cell medium or blood serum and reduce the undesirable effects of the Bys response induced by ionizing radiation (RIBE).

The Bys signal cascade induces an increase in micronuclei, SCE, mutations, apoptosis, protein formation associated with DNA damage, as well as decreased cell survival in cells receiving Bys signals. A strong relationship of these lesions with cell proliferation has been described in some Bys responses [57,58], in which increased Bys DNA damage is influenced by cell replication mechanisms and transcriptional activity [1,59,60].

Factors that may be involved in the induction of these Bys effects have been considered to include cytokines, RNA, TFG-β, TNF-α, IL-8, IL-6 and intermediate ROS produced by irradiated cells [1,17,18,19,20,21,51,52,53]. In different studies, it has been suggested that the sustained increase in ROS production in Bys cell populations contributes significantly to genotoxic damage by persistently inducing oxidative damage in DNA [61,62]. It has been suggested that the RIBE mechanism on DNA damage may be decisive due to the development of oxidative stress in non-irradiated recipient Bys cells, which leads to an increase the production of free radicals and inactivation of antioxidant defense mechanisms of cells [62,63,64,65,66]. Some authors have suggested that ROS could also be of great importance in the Bys effect, inducing a variety of signaling cascade effects [67].

In our study, we found that the Bys signaling cascade induces known effects in receptor cells: genotoxic effects expressed as a significant increase in micronuclei yield by in vivo and in vitro techniques, and cytotoxic effects expressed as a reduction in cell survival. However, the administration of RA that has genoprotective and radioprotective capacities in directly irradiated cells, lacks any genoprotective or radioprotective capacity in cells that receive Bys signals. Consistent with Harada et al. [68], radio-induced free radicals participate in signal formation in cells directly traversed by ionizing radiation but may not participate in Bys cells when they receive signals produced by irradiated cells.

However, another explanation could be offered for the inability of RA, an antioxidant agent, to protect cells that receive the cascade of signals from irradiated cells. Immediately after exposure to ionizing radiation, X-ray exposure produces a massive generation of reactive oxygen species (ROS)/free radicals in vivo. In general, ionizing radiation produces a large number of different radicals such as ^•^OH, e^−aq^ and H^•^ in the vicinity of DNA and its environs [33,55,69,70,71], which are mostly produced by the radiolysis water even in the absence of molecular oxygen. Its high reactivity produces an immediate reaction in the vicinity of its generation [41,55]. These ROS have a very short half-life and that can damage directly irradiated cells but cannot elicit damage to unirradiated cells [41]. However, when the generation of these ROS is massive, the damaging cytotoxic effect is no longer only localized in the immediate vicinity of the ionization, but can spread through reactive species and other radicals within the intracellular and even extracellular environment, increasing interaction with phospholipid cell structures and inducing lipid peroxidation processes that increase oxidative DNA damage at a greater distance and even a few hours after direct exposure to ionizing radiation has ended [71,72]. These free long-lived peroxides have a half-life of more than 24 h, so they have the possibility of traveling longer paths outside the irradiated cells and causing damage to non-irradiated cells. In this sense, it has been described that the administration of RA immediately after exposure to ionizing radiation significantly loses its genoprotective and cytoprotective abilities, precisely due to its inability to prevent the lipid peroxidation processes following exposure to ionizing radiation [28,32,71]. In this sense, Konopackac et al. show how ROS interceptors and in vitro production of peroxides (Vitamin C) reduce the frequency of chromosomal breaks, and their contributions to micronuclei production in non-irradiated receptor cells [14,64,65].

From a clinical point of view, RIBE modulation is a desirable target in radiotherapy due to its genotoxic potential that would allow the destruction of tumor cells and protect healthy cells that are in the same irradiation field. Radioprotective substances have been developed that protect against injuries caused by direct exposure to ionizing radiation, and other substances have also been developed that could reduce damage to receptor cells [1,28]. Early in vitro studies on RIBE have shown that some soluble antioxidants, ROS scavengers and nitric oxide (NO) radical scavengers, administered directly to cells prior to exposure to ionizing radiation, can diminish or nullify the signals that Bys effects produce on recipient cells [1,36]. Interestingly, NO radical scavengers have been shown to reduce Bys signal formation in irradiated cells but cannot protect bystander signal receptors from genotoxic effects [1,66]. Our study shows that some free radical scavenging antioxidants such as RA, with an important radioprotective capacity in irradiated cells, do not protect the receptor cells from Bys effects either.

## 5. Conclusions

Our results showed the loss of the in vivo and in vitro genoprotective and radioprotective capacities of RA in Bys cells when they receive the signals produced by irradiated cells, suggesting that only a part of ROS, the cytotoxic products of lipid peroxidation, participates in the lesions produced in the receptor Bys cells.

## Figures and Tables

**Figure 1 antioxidants-10-00231-f001:**
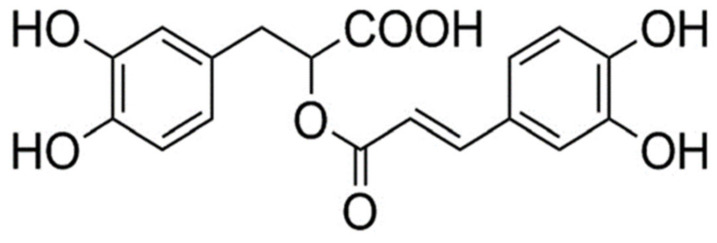
Chemical structure of rosmarinic acid (RA).

**Figure 2 antioxidants-10-00231-f002:**
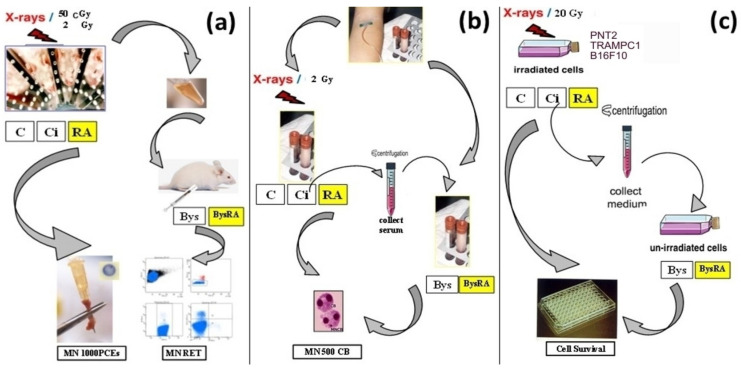
Models of radiation-induced bystander (Bys) effects (RIBE): (**a**) In vivo micronucleus assay (in mouse bone marrow (Micronuclei in polychromatic erythrocytes (MNPCEs)) and reticulocytes (Micronuclei in reticulocytes (MNRET))), (**b**) micronucleus test in vitro (Cytokinesis-block micronucleus assay (CBMN)), (**c**) MTT cytotoxicity assay.

**Figure 3 antioxidants-10-00231-f003:**
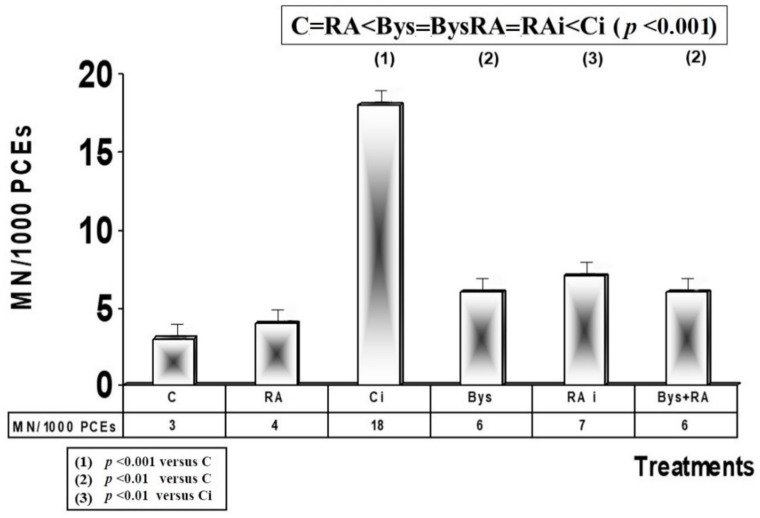
Micronuclei frequency in mouse bone marrow (C, control; RA, treated with rosmarinic acid; Ci, irradiated control; RAi, irradiated, previously treated with rosmarinic acid; Bys, treated with irradiated serum; Bys + RA, treated with irradiated serum, previously treated with rosmarinic acid).

**Figure 4 antioxidants-10-00231-f004:**
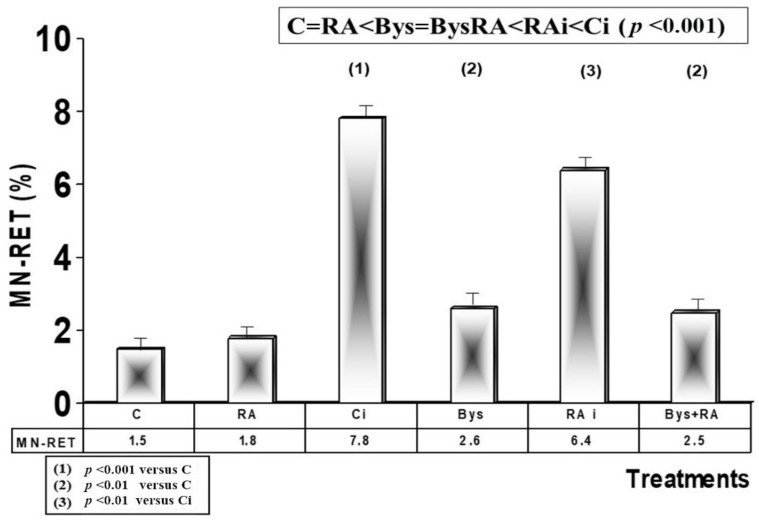
Micronuclei frequency in reticulocytes (C, control; RA, treated with rosmarinic acid; Ci, irradiated control; RAi, irradiated, previously treated with rosmarinic acid; Bys, treated with irradiated serum; Bys + RA, treated with irradiated serum, previously treated with rosmarinic acid).

**Figure 5 antioxidants-10-00231-f005:**
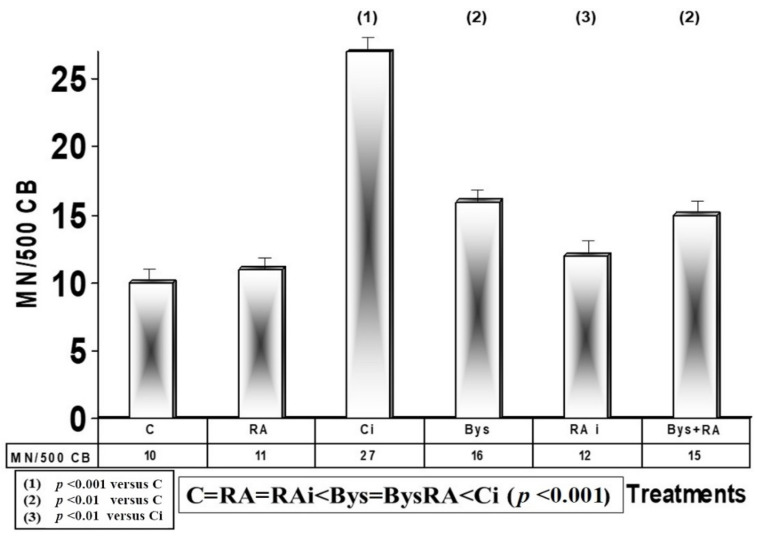
Micronucleus frequency in cytochalasin B-blocked irradiated human lymphocytes (C, control; RA, treated with rosmarinic acid; Ci, irradiated control; RAi, irradiated, previously treated with rosmarinic acid; Bys, treated with the irradiated serum; Bys + RA, treated with irradiated serum, previously treated with rosmarinic acid).

**Figure 6 antioxidants-10-00231-f006:**
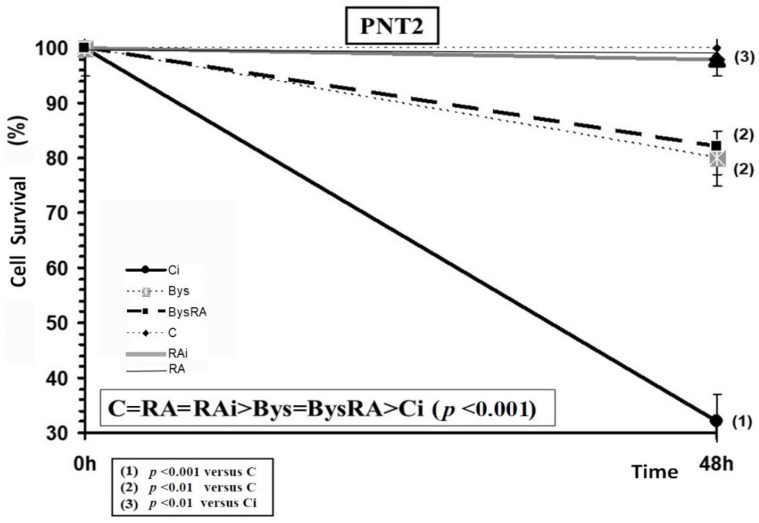
Cell survival of PNT2 cells irradiated to 20 Gy of X-rays after 48 h of incubation (C, control; RA, treated with rosmarinic acid; Ci, irradiated control; RAi, irradiated, previously treated with rosmarinic acid; Bys, treated with the medium of irradiated cell culture; BysRA, treated with the irradiated cell culture medium and with rosmarinic acid). Data are the mean and standard error of eight independent experiments.

**Figure 7 antioxidants-10-00231-f007:**
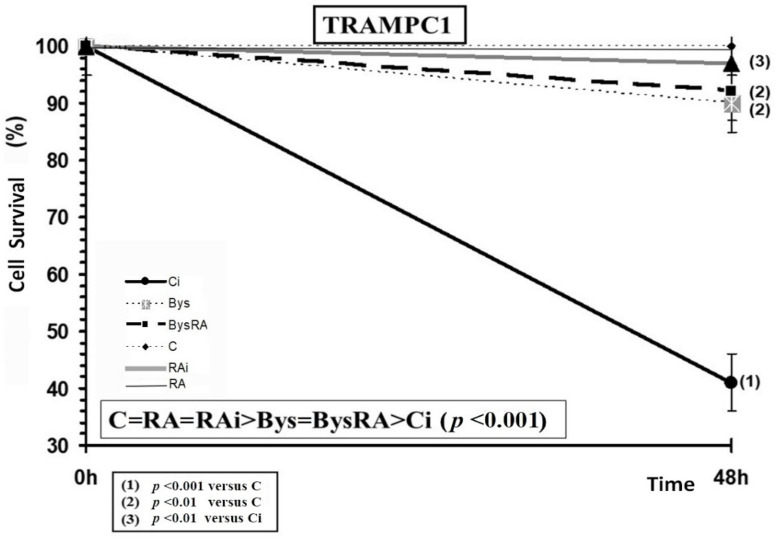
Cell survival of TRAMPC1 cells irradiated to 20 Gy of X-rays after 48 h of incubation (C, control; RA, treated with rosmarinic acid; Ci, irradiated control; RAi, irradiated, previously treated with rosmarinic acid; Bys, treated with irradiated cell culture medium; BysRA, treated with irradiated cell culture medium and with rosmarinic acid). Data are the mean and standard error of eight independent experiments.

**Figure 8 antioxidants-10-00231-f008:**
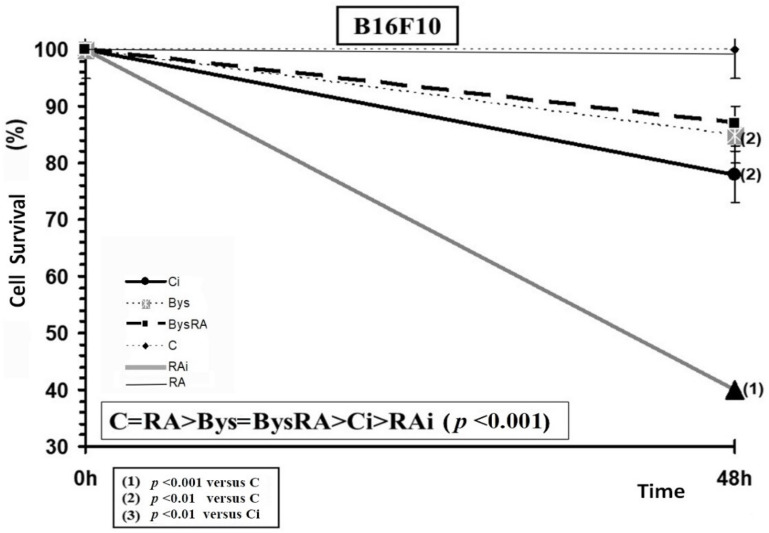
Cell survival of B16F10 cells irradiated to 20 Gy of X-rays after 48 h of incubation (C, control; RA, treated with rosmarinic acid; Ci, irradiated control; RAi, irradiated, previously treated with rosmarinic acid; Bys, treated with the medium of irradiated cell culture; BysRA, treated with the irradiated cell culture medium and rosmarinic acid). Data are the mean and standard error of eight independent experiments.

**Figure 9 antioxidants-10-00231-f009:**
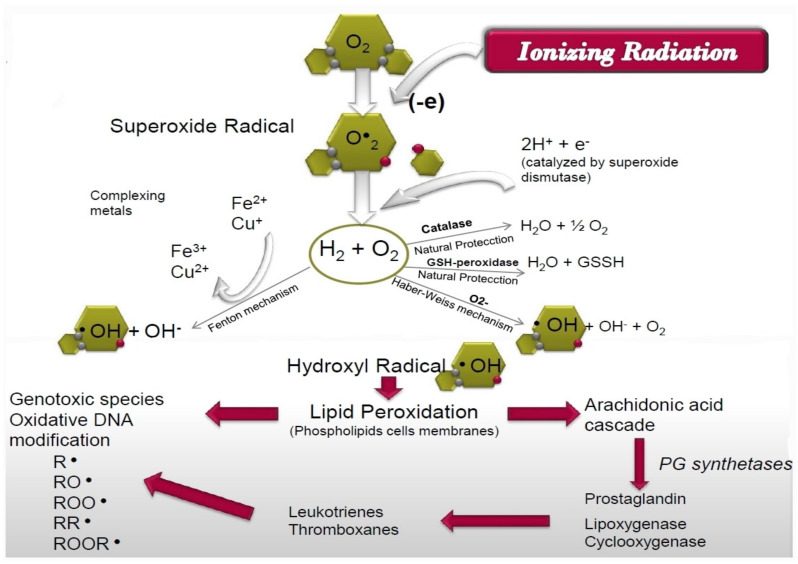
Schematic representation of the cellular injury mechanism of ionizing radiation-induced damage.

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
