# Peer review of "Radiation-Induced Bystander Effect: Loss of Radioprotective Capacity of Rosmarinic Acid In Vivo and In Vitro"

_antioxidants, 2021, doi:10.3390/antiox10020231_

Round 1
Reviewer 1 Report
Olivares et al. have presented an interesting manuscript that explores the effect of antioxidant rosmarinic acid (RA) on several different irradiated and bystander cell populations, from in vitro and in vivo cell populations. Genotoxic and cytotoxic endpoints were measured. The study demonstrated that while RA substantially reduced cellular damage in irradiated cells, exposure to RA conferred no significant effects on bystander populations. It was hypothesised that RA’s mechanism of action led to reduced adverse effects of short-lived ROS, rather than longer-lived ROS that would remain present to confer damage to bystander cultures.
The study is generally adequately designed and clearly presented, although there were several aspects that should be explained more explicitly to the reader or that require further consideration. It would be useful to include a justification of why different techniques to assess genotoxicity and cytotoxicity were used for different cell types. For example, flow cytometry and microscopy analyses were used to assess micronucleus frequency, in different cell types. At first read, this appears somewhat inconsistent and it would therefore be useful for the authors to explain why different techniques were used. Micronucleus analysis was performed for some cell types and not others; for example, the three cell lines were only used for the MTT assay and not micronucleus assessment. It would again be useful to justify this discrepancy. If there is not a strong justification, then it is recommended that authors complete additional experiments to collect micronucleus data for the cell lines.
It is usually recommended that cytotoxicity data are obtained in parallel with genotoxicity data, in order to confirm that the genotoxic effects observed were not merely owing to secondary toxicity rather than direct effects of radiation, etc. A justification of why cytotoxicity was not studied alongside is needed, particularly for cultured cells.
Further minor points for improving the manuscript are included below:
Abstract: The abstract is slightly confusing to read in places. It should begin with a general introductory sentence on the background of the study. In Line 16, it is stated that the cell lines used have “different degrees of radiosensitivities”. It would be useful to revisit and support this claim later in the manuscript, providing evidence from the literature. In Line 19, the authors should make it clearer earlier in the sentence that irradiated cells are being referred to, in terms of RA’s radioprotective effect. In Line 24, please replace “a part of these” with a more specific term such as “certain subtypes of”.
The Introduction is concise. It would however benefit from further justification of the medical importance of understanding bystander effects. More specific information on genotoxicity and cytotoxicity mechanisms underlying bystander effects within the current literature would be welcomed, although this is covered in some detail in the Discussion. The Discussion is considerably longer than the Introduction and the authors might consider moving some general background information to the Introduction and referring back to this in the Discussion in the context of their own results.
The manuscript contains some spelling and grammatical errors; proof-reading is recommended. For example, in line 143, “week” should be “weeks”.
Line 106 – Add that these were human donors to make this clear.
Line 154 – Perhaps it could be mentioned here that the study included a non-irradiated control + RA, for completion.
Line 274 – Translate part of heading to English?
Author Response
Comment. Olivares et al. have presented an interesting manuscript that explores the effect of antioxidant rosmarinic acid (RA) on several different irradiated and bystander cell populations, from in vitro and in vivo cell populations. Genotoxic and cytotoxic endpoints were measured. The study demonstrated that while RA substantially reduced cellular damage in irradiated cells, exposure to RA conferred no significant effects on bystander populations. It was hypothesised that RA’s mechanism of action led to reduced adverse effects of short-lived ROS, rather than longer-lived ROS that would remain present to confer damage to bystander cultures.
The study is generally adequately designed and clearly presented, although there were several aspects that should be explained more explicitly to the reader or that require further consideration. It would be useful to include a justification of why different techniques to assess genotoxicity and cytotoxicity were used for different cell types. For example, flow cytometry and microscopy analyses were used to assess micronucleus frequency, in different cell types. At first read, this appears somewhat inconsistent and it would therefore be useful for the authors to explain why different techniques were used. Micronucleus analysis was performed for some cell types and not others; for example, the three cell lines were only used for the MTT assay and not micronucleus assessment. It would again be useful to justify this discrepancy. If there is not a strong justification, then it is recommended that authors complete additional experiments to collect micronucleus data for the cell lines.
It is usually recommended that cytotoxicity data are obtained in parallel with genotoxicity data, in order to confirm that the genotoxic effects observed were not merely owing to secondary toxicity rather than direct effects of radiation, etc. A justification of why cytotoxicity was not studied alongside is needed, particularly for cultured cells.
Response. Genotoxicity tests are designed to detect compounds or physical agents that directly or indirectly induce damage to the genetic material by different mechanisms, and constitute a fundamental requirement for the evaluation of the inherent toxicological or mutagenic characteristics of physical or chemical agents. Studies involving genetic mutations may be carried out through different in vitro and in vivo tests, of variable complexities, depending on the biological mechanism to be analyzed.
Currently, there is no single mutagenesis test capable of solely detecting all types of mutations, therefore, regulatory agencies require a battery of genotoxicity studies on all new drugs before their approval.
Different genotoxicity tests can be performed either in vitro or in vivo. These tests are regulated and systematized so that their results will be reliable and comparable Among those currently used are the Ames test, mammalian chromosomal aberration test, Comet test or single-cell electrophoresis technique, mammalian erythrocyte micronucleus test, in vivo cytogenetic tests, and the cytokinesis-block micronucleus test on human lymphocytes We used the micronucleus analysis because it is has been recommended by the IAEA (40) for screening of the genoprotective capacities of chemical substances against injuries induced by exposure to ionizing radiation. Therefore, we studied the genoprotective effect of RA through both in vitro and in vivo micronucleus analysis. However, the in vitro MN test in irradiated human lymphocytes with cytokinetic blockade with Cytochalasin-B (CBMN) is possibly the most used for the analysis of mutagenic lesions induced by ionizing radiation. Following the recommendations to confirm the results through different tests, we used the in vivo MN test in polychromatophilic erythrocytes from mouse bone marrow, which also reveals the absorption, distribution and bioavailability of RA in live irradiated animals that can demonstrate the effect of RA and have no influence on the in vitro CBMN assay. Although the MN assay in mouse bone marrow described in 1975 is still widely used as per the FDA recommendation, in recent years the reticulocyte micronucleus test which uses flow cytometry has been described. This method allows for the differentiation the different phases of blood reticulocytes (which correspond to polychromatophilic erythroblasts as they exit the bone marrow into peripheral blood) and provide a possibly for more accurate analysis of ionizing radiation-induced lesions; analyzing hundreds of thousands of cells in each determination. Following the criteria of the regulatory agencies, the homogeneity nature of the tests used allows confirmation of the results obtained from both the damaging ability of ionizing radiation and the genoprotective capacity of RA.
On the other hand, to measure the inhibition of cell growth or cell survival as used in radiobiology, we used the MTT assay based on the ability of functionally active mitochondria to convert MTT into formazan, the amount of which is directly proportional to the number of living cells. We selected three cell lines characterized by their different degree of radiosensitivity: normal epithelial cells of the human prostate considered very radiosensitive and those of murine melanoma B16F10 considered highly radioresistant. Given that one group are normal cells and other are tumor cells, we also included transgenic prostate tumor cells which maintains the radiosensitivity characteristics of the prostate. With these we tried to determine the cytotoxic capacity of ionizing radiation by determining the radiation-induced cell death and, where appropriate, we used the increase in cell survival produced by RA as an expression of its radioprotective capacity.
For all the above, it was of no interest to carry out a study of micronuclei in cell cultures of PNT2, B16F10 or TRAMPC1 cells, since the results will be of no value in interpreting a possible antimutagenic or genoprotective effect of ionizing radiation or of genoprotective capacity. These micronucleus assays on PNT2, TRAMP-C1, and B16F10 have not been internationally validated for this purpose and are of no utility.
Similarly, it is not possible to perform a study of cytotoxicity or inhibition of cell growth in cells that never divide such as blood reticulocytes, bone marrow erythroblasts, or in cells that are blocked in their first mitotic division (CBMN). For this reason, although the tests are found in different cell types, they correspond to specific tests that are adequate to determine the genoprotective and cytoprotective capacities respectively. By and large, they show both the damaging capacity of ionizing radiation and the protective capacity of RA in irradiated cells and in bystander cells.
At all times, the secondary toxicity produced by the concentrations of RA administered has been determined by the control groups so that it does not interfere with the results obtained
To clarify these aspects, we have included the following paragraphs in Discussion (lines 748-785):
“We used three complementary cytogenetic assays to evaluate the mutagenic / genotoxic capacity of ionizing radiation by determining the frequency of appearance of micronuclei in cells and animals exposed to ionizing radiation.
Similarly, a reduction in the frequency of occurrence of these micronuclei makes it possible to determine the protection capacity (antimutagenic effect) of RA against chromosomal damage induced by ionizing radiation. Since there is no single test that is capable of detecting all genotoxic end points, the complementary use of several genotoxicity tests is recommended. The homogeneity of results allows for the assessment of the mutagenic capacities of ionizing radiation, irradiated culture medium and irradiated blood serum, as well as the possible antimutagenic capacity of RA. In this study we applied the most widely used micronucleus assays to determine the genotoxic capacity of ionizing radiation both in vitro and in vivo which are considered universally validated, technologically accessible and useful for evaluating genetic instability induced by genotoxic agents and especially in screening for radioprotective substances (IAEA). In the in vitro test on irradiated human lymphocytes blocked with cytochalasin B, the micronucleus, a chromosome fragment that remains outside the mitotic spindle after the first cell division is observed as a cytoplasmic body with nuclear characteristics that correspond to genetic material not incorporated correctly to daughter cells. This reflects a chromosomal aberration produced by chromosomal breaks, by errors during DNA replication and subsequent cell division and / or by exposure to genotoxic agents or substances. In the in vivo test, the absorption and bioavailability of the physical and chemical agents tested may alter the expected results. In these in vivo tests, the micronucleus is also a chromosomal fragment with nuclear characteristics that remains inside the polychromatophilic erythroblasts of bone marrow (mouse bone marrow). It may also remain inside the circulating reticulocytes in the blood showing an inexplicable delay in its elimination from the cell as an expression of the delay in cell maturation and proportional to the chromosomal damage induced by a chemical or physical agent.
In this sense, our genotoxicity tests show a high genotoxic capacity both X-rays with which was used to directly irradiate the cells and animals studied, as well as the irradiated culture medium and the serum obtained from irradiated animals.
However, we observed significant differences between the two since the serum from irradiated animals and the irradiated culture medium expressed significantly lower mutagenic capacities than direct exposure to X-rays.
On the other hand, the reduction of MN after the administration of RA allows for the quantification of the antimutagenic capacity of RA compared to direct exposure to X-rays, while the lack of effect on the frequency of appearance of MN in all the tests carried out expresses the absence of genoprotective capacity of RA in cells that receive Bys signals from irradiated serum or culture medium.”
Comment. Further minor points for improving the manuscript are included below:
Abstract: The abstract is slightly confusing to read in places. It should begin with a general introductory sentence on the background of the study. In Line 16, it is stated that the cell lines used have “different degrees of radiosensitivities”. It would be useful to revisit and support this claim later in the manuscript, providing evidence from the literature. In Line 19, the authors should make it clearer earlier in the sentence that irradiated cells are being referred to, in terms of RA’s radioprotective effect. In Line 24, please replace “a part of these” with a more specific term such as “certain subtypes of”.
Response. We have modified the abstract following your instructions (lines 12-25):
“In radiation oncology the modulation of the bystander effect is a target both for the destruction of tumor cells and to protect healthy cells. With this objective we determine whether the radioprotective capacity of rosmarinic acid (RA) can affect the intensity of these effects. Genoprotective capacity was obtained by determining the micronuclei frequencies in vivo and in vitro assays and the cell survival was determined by the MTT assay in three cell lines (PNT2, TRAMPC1 and B16F10) both in direct exposure to X-rays and after the production of radiation-induced bystander effect. The administration of RA in irradiated cells produced a decrease in the frequency of micronuclei both in vivo and in vitro, and an increase in cell survival, as expression of its radioprotective effect (p <0.001) attributable to its ability to scavenge radioinduced free radicals (ROS). However, RA does not achieve any modification in the animals receiving serum or in the cultures treated with the irradiated medium, which expresses an absence of radioprotective capacity. The results suggest that ROS participates in the formation of signals in directly irradiated cells, but only certain subtypes of ROS, the cytotoxic products of lipid peroxidation, participates in the creation of lesions in recipient cells.”
We have modified in Material and Method point 2.3. MTT Cytotoxicity Assay / Cell culture. The paragraph now reads (lines 271-289):
“We selected three cell lines based on their degree of radiosensitivity. On the one hand, melanoma cells are characterized by high radioresistance (Hall), which is why we used murine metastatic melanoma cells B16F10 which were kindly donated by Dr. Hearing from the National Cancer Institute (Bethesda, MA, USA). In contrast, human prostate cells (Hall) are traditionally considered radiosensitive so we selected human prostate epithelial cells PNT2 cells which were obtained from the European Collection of Cell Cultures (ECACC), Health Protection Agency Culture Collection (catalog n: 95012613; UK). Finally, to examine the tumor influence on these radiosensitive cells we also used Transgenic adenocarcinoma of mouse prostate (TRAMP-C1) cells which were obtained from the American Type Culture Collection (catalog n: CRL-2730, USA). The PNT2 cells were cultured in RPMI-1640 medium supplemented with 10% fetal bovine serum, glutamine (2 mM) containing streptomycin 100 μg/ml and 100 IU/ml penicillin. Both TRAMP-C1 and B16F10 cells were cultured in Dulbecco’s Modified Eagle’s Medium and Nutrient Mixture-F2 Ham (DMEM)/F12K (1:1). Both mediums contained 10% fetal bovine serum (Gibco, BRL, Louisville, KY, USA) and 5% penicillin/streptomycin. The cell cultures were maintained at 37˚C and a relative humidity of 95%, in an atmosphere of 5% CO2. Rosmarinic acid were administered by dissolving it in phosphate-buffered saline (PBS) to a concentration of 25 µM and consistently adding 25 μl of the solution to each well immediately before X-irradiation (5-10 mn). At all times the cell cultures were kept free of mycoplasma spp.”
Comment. The Introduction is concise. It would however benefit from further justification of the medical importance of understanding bystander effects. More specific information on genotoxicity and cytotoxicity mechanisms underlying bystander effects within the current literature would be welcomed, although this is covered in some detail in the Discussion. The Discussion is considerably longer than the Introduction and the authors might consider moving some general background information to the Introduction and referring back to this in the Discussion in the context of their own results.
Response. We have moved Two paragraphs from the Discussion to the Introduction. These paragraphs are:
-(lines 81-89)
“It can be assumed that the initial trigger for bystander responses is a nonspecific activation of a redox-sensitive signaling pathways through mitochondrial induction of ROS/ RNS [35]. ROS induced by ionizing radiation are considered to be the initiator, and nitric oxide the effector that activates the bystander process through a signal transduction pathways [39]. TGF-beta protein has been described as an important agent in signal activation and in amplifying the response to oxidative stress [42, 43, 44] which together with other inflammatory cytokines appear to increase in irradiated tissues [45,46] and contribute to injury induced by ionizing radiation and also in RIBE [35,47,48].”
-(lines 99-108)
“From a clinical point of view, RIBE modulation is a desirable target in radiotherapy due to its genotoxic potential that would allow the destruction of tumor cells and protect healthy cells that are in the same irradiation field. Radioprotective substances have been developed that protect against injuries caused by direct exposure to ionizing radiation, and other substances have also been developed that could reduce damage to receptor cells [21,35]. In radiation oncology, radioprotective substances could protect the normal tissue surrounding tumors and ameliorate the side effects of therapy, but, in this setting, care must be taken not to diminish the destruction of tumor cells. Therefore, a complex un-der-standing of passerby signaling pathways and potential molecular targets is crucial [74,75].”
Comment. The manuscript contains some spelling and grammatical errors; proof-reading is recommended. For example, in line 143, “week” should be “weeks”.
Response. This has been corrected
Comment. Line 106 – Add that these were human donors to make this clear.
Response. The paragraph has been modified. Now it reads (lines 165-174): “For the in vitro genotoxicity test, 20 ml of venous blood was obtained from the elbow flexure from three apparently healthy human donors after their informed consent, which were collected in heparinized tubes for the performance of the human lymphocyte micronucleus assay with cytokinesis-block (CBMN) as previously described [30] and adapted by the International Atomic Energy Agency for the screening of genoprotective substances against damage induced by ionizing radiation [31]. In the RA group, twenty-five microliters (25 µl) of a 25 µM concentration of RA dissolved in PBS were added to 2 ml of heparinized human blood (non-irradiated control, RA) and immediately prior to X-ray exposure (RAi). The number of micronuclei in at least 3,000 CB cells for each treatment was determined by three specialists who analyzed the slides using light microscopes in a double-blind study.”
Comment. Line 154 – Perhaps it could be mentioned here that the study included a non-irradiated control + RA, for completion.
Response. We have modified this paragraph to read (line 313): “At the time of starting X-ray exposure, we changed the medium to fresh medium in all cells (non-irradiated cells, non-irradiated cells + RA, irradiated cells and irradiated cells + RA); 4 h after irradiation, this medium was collected, where the factors that can induce bystander effects are found.”
Comment. Line 274 – Translate part of heading to English?
Response. This has been modified, it now reads (line 586): “MTT Cytotoxicity Assay”.
Reviewer 2 Report
Many investigators found that non-irradiated cells suffer the signals of cell damage from irradiated cells, this effect is called radiation-induced bystander effect (RIBE). RIBE is potential risk indicator for radiation carcinogenesis, but the mechanisms are not fully understood. Previous studies indicated that rosmarinic acid possesses radioprotective and antioxidant capabilities through serving as ROS scavenge. The author proposed whether rosmarinic acid could also attenuate RIBE after radiotherapy, and further expect to clarify the mechanisms of RIBE. However, the results showed that rosmarinic acid didn’t reduced RIBE. In this study, rosmarinic acid is proved an excellent antioxidant to direct irradiation but it is ineffective to decrease the damage induced from RIBE. Although the purpose of this manuscript is clear, however, the study design is too sketchy. In addition, there are still some questions to be corrected and explained.
Introduction
- The full name of Bys has provided in first mention in manuscript and followed by abbreviations.
- I think that the content of Reference 14 & 15 is not associated with bystander effect. Please explain it.
Materials and Methods
In vivo micronucleus assay:
Why did not use gavage needles to feed rosmarinic acid? This can ensure that each mouse eat the same amount of rosmarinic acid.
In vitro micronucleus assay:
- Please uniform the abbreviation of cytokinesis-block micronucleus. (MNCB or CBMN?)
- When the addition of rosmarinic acid before radiation?
MTT cytotoxicity assay:
When the addition of rosmarinic acid before X ray-irradiation? Please describe it.
Results
Major concern:
- All experimental design for verifying argument is too simple.
For example, MTT assay is not standard to evaluate cytotoxicity and cell viability in radiation treatment. The authors must provide clonogenic assay and annexin V-PI apoptosis assay.
In addition, the DNA damage response signaling pathways must be assessed for genotoxic effect.
- I want to know whether the serum and culture medium from Rai group can enhance the levels of MNPCE, MN-RET, CBMN, and cytotoxicity of untreated group. This is better evidence to prove RA can’t affect Bys.
Minor concern:
- In figure 5. The experimental procedure of Bys and Bys+RA group must write in Materials and Methods.
- Figure 6~8. There is no data of RA alone (only description in content), so it is inappropriate to show C=RA
Discussion
- The addition of RA has a paradoxical result in B16F10. Is there other reason besides RA decreased glutathione?
- The factor involved in the induction of Bys is not mentioned in Reference 58~60. Please correct it.
Author Response
Comment. Many investigators found that non-irradiated cells suffer the signals of cell damage from irradiated cells, this effect is called radiation-induced bystander effect (RIBE). RIBE is potential risk indicator for radiation carcinogenesis, but the mechanisms are not fully understood. Previous studies indicated that rosmarinic acid possesses radioprotective and antioxidant capabilities through serving as ROS scavenge. The author proposed whether rosmarinic acid could also attenuate RIBE after radiotherapy, and further expect to clarify the mechanisms of RIBE. However, the results showed that rosmarinic acid didn’t reduced RIBE. In this study, rosmarinic acid is proved an excellent antioxidant to direct irradiation but it is ineffective to decrease the damage induced from RIBE. Although the purpose of this manuscript is clear, however, the study design is too sketchy. In addition, there are still some questions to be corrected and explained.
Introduction
The full name of Bys has provided in first mention in manuscript and followed by abbreviations.
Response. Thanks, we have used this abbreviation throughout the manuscript
Comment. I think that the content of Reference 14 & 15 is not associated with bystander effect. Please explain it.
Response. We have removed these references. They correspond to the first mechanisms that were invoked to explain the Bystander effect, but they do not allign with the paragraph to which they are attached.
Comment.Materials and Methods
In vivo micronucleus assay:
Why did not use gavage needles to feed rosmarinic acid? This can ensure that each mouse eat the same amount of rosmarinic acid.
Response. In different publications we used three different routes for the administration of different substances before exposure to ionizing radiation, including RA:
- Administration of 0.2 ml of RA solution at a concentration of 180 mg / ml in saline by nasogastric tube
- Intraperitoneal injection of RA at a concentration of 150 mg / ml in PBS.
- Oral administration in 0.2% RA in drinking water for 4 and / or 7 days.
We have not yet established significant differences between the three routes of administration. In addition, the consistency of the results in the oral route of administration coerced us to select this route of administration based on the high solubility of RA in water coupled with its ease of administration and the ethics approval.
Comment. In vitro micronucleus assay:
Please uniform the abbreviation of cytokinesis-block micronucleus. (MNCB or CBMN?)
Response. We have standardized the abbreviation. CBMN
Comment. When the addition of rosmarinic acid before radiation?
Response. Always before exposure to ionizing radiation. When administered 5-10 min after exposure to ionizing radiation, the genoprotective effect is significantly reduced.
We have included this observation in the lines 143, 172 and 288.
Comment. MTT cytotoxicity assay:
When the addition of rosmarinic acid before X ray-irradiation? Please describe it.
Response. Immediately before exposure to ionizing radiation. Always within 10 minutes prior to exposure to ionizing radiation.
We have included this observation (lines 286-289): “Rosmarinic acid were administered by dissolving it in phosphate-buffered saline (PBS) to a concentration of 25 µM and consistently adding 25 μl of the solution to each well immediately before X-irradiation (10 mn)”.
Comment. Results
Major concern:
All experimental design for verifying argument is too simple.
For example, MTT assay is not standard to evaluate cytotoxicity and cell viability in radiation treatment. The authors must provide clonogenic assay and annexin V-PI apoptosis assay.
Response. Currently, the clonogenic assay is the most widely used to determine cell viability, although MTT has also been used for many years to determine the inhibition of cell growth. The following paragraph has been included in the discussion (line 807):
“A limitation of the study is the exact determination of cell survival using the MTT technique. Currently the clonogenic assay and the annexin V-PI apoptosis assay are the standard techniques for evaluating cell viability in ionizing radiation treatments. However, the comparative analysis of the results obtained with the MTT test makes it possible to assess the effect of RA in both directly irradiated cells and in Bys cells”.
Comment. In addition, the DNA damage response signaling pathways must be assessed for genotoxic effect.
Response. We have included the following paragraphs into the discussion:
-(lines 724-746)
“In the "target" theory, exposure to ionizing radiation increases cellular oxidation pro-cesses causing molecular oxygen to give rise to superoxide radicals (O●2), hydrogen per-oxide and hydroxyl radicals (●OH), successively [41,55]. Of special interest is the intense generation of hydroxyl radicals since they are presently considered to have the highest cytotoxic capacity an adverse effect that can possibly be mitigated by the administration of radioprotective substances such as RA. These hydroxyl radicals have high chemical reac-tivity and an estimated half-life of 10-9 s, which in practice implies an immediate reaction close to the place where they are generated. However, when the generation of these hy-droxyl radicals is massive, they interact with the cellular phospholipoid structures in-ducing the lipoperoxidation processes and gradually producing lipoperoxy radicals (R●, RO●, RR●, ROO●, ROOR●). An escalation of these free radical accumulation over time in-creases cellular genotoxic and cytotoxic effects, in what has been considered a delayed secondary reaction produced by ionizing radiation which prolongs the cytotoxic effect for at least 24 hours after the exposure to ionizing radiation has ended. On the other hand, this increase in lipid peroxidation increases lipoxygenase, cycloxygenase and phospho-lipase activities which leads to an increase the secretion of lysosomal enzymes and the re-lease of arachidonic acid from cell membranes that also increase the intensity of cellular inflammatory response. Through this delayed reaction of lipoperoxy radicals, ionizing radiation can induce effects at some distance from the place where the exposure to ioniz-ing radiation initially occurred and for a much longer time [28,41,55,56] (Figure 9).
Round 2
Reviewer 1 Report
I thank the authors for clarifying the points raised, including their justification of their chosen methodology. I now consider the manuscript to be suitable for publication.
Reviewer 2 Report
The authors have addressed the comments from the reviewers.